# Bipolar Self-attention for Spiking Transformers

**Shuai Wang**[1], **Malu Zhang**[1,2*] **Jingya Wang**[1], **Dehao Zhang**[1], **Yimeng Shan**[1], **Jieyuan Zhang**[1],
**Yichen Xiao**[1], **Honglin Cao**[1], **Haonan Zhang**[1], **Zeyu Ma**[1], **Yang Yang**[1], **Haizhou Li**[2,3]

[1]University of Electronic Science and Technology of China

[2]Shenzhen Loop Area Institute, [3]The Chinese University of Hong Kong (Shenzhen)

## Abstract

Harnessing the event-driven characteristic, Spiking Neural Networks (SNNs) present a promising avenue toward energy-efficient Transformer architectures. However, existing Spiking Transformers still suffer significant performance gaps compared to their Artificial Neural Network counterparts. Through comprehensive analysis, we attribute this gap to these two factors. First, the binary nature of spike trains limits Spiking Self-attention (SSA)'s capacity to capture negative–negative and positive–negative membrane potential interactions on Querys and Keys. Second, SSA typically omits Softmax functions to avoid energy-intensive multiply-accumulate operations, thereby failing to maintain row-stochasticity constraints on attention scores. To address these issues, we propose a Bipolar Self-attention (BSA) paradigm, effectively modeling multi-polar membrane potential interactions with a fully spike-driven characteristic. Specifically, we demonstrate that ternary matrix multiplication provides a closer approximation to real-valued computation on both distribution and local correlation, enabling clear differentiation between homopolar and heteropolar interactions. Moreover, we propose a shift-based Softmax approximation named Shiftmax, which efficiently achieves low-entropy activation and partly maintains row-stochasticity without non-linear operation, enabling precise attention allocation. Extensive experiments show that BSA achieves substantial performance improvements across various tasks, including image classification, semantic segmentation, and event-based tracking. These results establish its potential as a fundamental building block for energy-efficient Spiking Transformers.

## 1 Introduction

As the core computational unit of Transformers [5, 15, 11, 32, 9], self-attention mechanism dynamically models the global dependencies among sequence elements, overcoming the long-range dependency challenges [33, 10]. However, its' $O(N^2d)$ computational complexity [18, 48] incurs an exponential rise during both training and inference, restricting its application in many resource-constrained environments. Consequently, how to develop energy-efficient and high-performance Transformers remains a critical research focus.

Spiking Neural Networks (SNNs) [23, 12] have gained significant attention due to their brain-inspired dynamics [17, 24]. Spiking neurons fire discrete spikes only when activated, remaining silent otherwise. Compared with Artificial Neural Networks (ANNs) that rely on multiply-accumulate (MAC) computation, the spike-driven mechanism [61, 43, 44] in SNNs supports sparse accumulate (AC) operations [2]. Such sparse spike-based computation [20, 56] delivers significant power efficiency, particularly on neuromorphic platforms such as Tianjic [26, 8, 22] and Loihi [6, 25]. Recently, numerous researchers focus on developing Spiking Transformers, including Spikformer [70], Spikingformer [67], Spike-driven Transformers [51, 49, 52], SpikingResformer [30], and QKformer [68].

---

*Corresponding author: maluzhang@uestc.edu.cn

These approaches enhance the performance ceiling of SNNs in various tasks [64, 45, 36, 31, 62, 34], demonstrating that Spiking Transformers achieve a trade-off between high performance and efficiency.

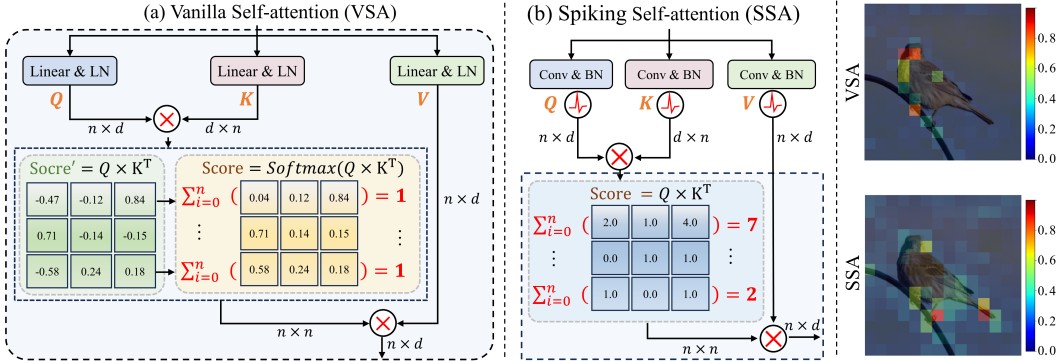

Figure 1: Comparison of Vanilla Self-attention (VSA) and Spiking Self-attention (SSA) mechanisms. **Score′** contains rich polarity information and **Score** maintains strict row-stochasticity via Softmax in VSA, while **Score** in SSA lacks negative polarity and row-stochasticity constraints.

As a core component of Spiking Transformer architectures, the Spiking Self-ttention (SSA) [70, 51] computational paradigm represents the critical factor limiting their performance ceiling. As shown in Fig.1, Vanilla Self-attention (VSA) in ANNs effectively captures magnitude and multi-polarity information by computing **Score′** through query (Q) and key (K) correlation. Meanwhile, it calculates the **Score** using the Softmax function under row-stochasticity constraint. In contrast, SSA in SNNs operates with binary spike trains (0 or 1), which not only sacrifices quantization precision but also disregards negative-negative and positive-negative interactions. Furthermore, to maintain energy efficiency, SSA typically omits Softmax function, resulting in severely imbalanced row-stochasticity in **Score**. These issues limit SSA's ability to effectively compute attention allocation between Q and K, causing SSA more like a simplified Token Mixer [54, 55]. Therefore, how to overcome these limitations is crucial for pushing Spiking Transformers beyond their current performance bottlenecks.

In this paper, we propose a Bipolar Self-attention (BSA) paradigm to effectively address these issues. Unlike the SSA paradigm that exclusively captures positive-positive Q-K interactions, BSA employs ternary spiking neurons [14, 35] to comprehensively process both homopolar and heteropolar interaction patterns in Q-K correlation computation. We theoretically demonstrate that ternary matrix multiplication more closely approximates real-valued computation in terms of both distributional similarity and local correlation than binary part. Moreover, we propose the innovative Shiftmax method, which approximates Softmax's low-entropy activation characteristics and row-stochasticity constraints through energy-efficient bit-shift operations. Finally, we conduct extensive experiments across diverse tasks including image classification [70, 68], semantic segmentation [49, 52], and event-based tracking [58, 28], consistently demonstrating that BSA delivers substantial performance improvements. The main contributions of our work are outlined as follows:

- We first identify two limitations of Spiking Self-attention: (1) binary matrix products exclusively capture positive-positive correlations while neglecting negative–negative and positive–negative polarity features, leading to a complete loss of polarity information. (2) without Softmax, attention scores across different rows exist on incomparable scales, rendering attention allocation ineffective. These deficiencies prevent SSA from fully harnessing the potential of the self-attention mechanism.

- We propose Bipolar Self-attention (BSA) to overcome these limitations. BSA employs ternary matrix products to extract Q-K correlations, comprehensively process different polarity interaction patterns. In addition, we propose a Shiftmax method to approximate Softmax that achieves low-entropy activation and maintains partial row-stochasticity without non-linear operations, enabling precise attention allocation.

- Extensive experiments demonstrate that BSA achieves significant performance improvements across various advanced Spiking Transformers on ImageNet-1K. Furthermore, our method establishes state-of-the-art performance in both semantic segmentation and event-based tracking tasks compared to existing SNNs approaches.

## 2 Related Work

Recently, growing attention has been paid to energy-efficient Spiking Transformers [4, 50]. For instance, Spikformer [70] first proposes Spiking Self-attention computation, establishing the first Spiking Transformer. Building on this, Spikingformer [67] introduces a hardware-friendly residual learning architecture that avoids non-spike computations in SNNs. Then, the Spike-driven Transformers (SDT-V1) [51] incorporates the Hadamard product into the self-attention module, achieving a fully spike-driven mechanism. Furthermore, SpikingResformer [30] integrates a Dual Spike self-attention module, enhancing both performance and energy efficiency. Recently, QKformer [68] and Spike-driven Transformer-V3 [52] both significantly elevate the performance ceiling of Spiking Transformers. Nonetheless, these models primarily utilize self-attention as a token mixer [54], without paying attention to exploring effective similarity calculations suited to spike trains [60, 1]. Therefore, designing an innovative spiking self-attention mechanism that fully leverages self-correlation computation under spike-driven characteristics is crucial for further advancement.

## 3 Preliminary

### 3.1 Vanilla Self-attention

The self-attention mechanism effectively captures global dependencies within sequences by dynamically allocating attention across tokens [33, 29, 13]. Given an input matrix $\mathbf{X} \in \mathbb{R}^{n \times d}$, where $n$ denotes sequence length and $d$ represents embedding dimensionality, the mechanism initially projects $\mathbf{X}$ into query $\mathbf{Q}$, key $\mathbf{K}$, and value $\mathbf{V}$ matrices via distinct parameter matrices $\mathbf{W}_Q$, $\mathbf{W}_K$, and $\mathbf{W}_V$:

$$\tilde{\mathbf{X}} = \text{LN}(\mathbf{X}), \qquad \mathbf{Q} = \text{Linear}(\tilde{\mathbf{X}}), \quad \mathbf{K} = \text{Linear}(\tilde{\mathbf{X}}), \quad \mathbf{V} = \text{Linear}(\tilde{\mathbf{X}}). \tag{1}$$

Here, LN denotes layer normalization, and Linear refers to a fully connected layer. To determine contextual relationships, the model computes token-wise similarities through dot-product operations between $\mathbf{Q}$ and $\mathbf{K}$, generating an attention score matrix:

$$\mathbf{Score}' = \mathbf{Q} \times \mathbf{K}^\top, \quad \mathbf{Score} = \text{Softmax}\left(\frac{\mathbf{Score}'}{\sqrt{d}}\right), \quad \mathbf{Attn} = \mathbf{Score} \times \mathbf{V}. \tag{2}$$

The mechanism applies a scaling factor $\sqrt{d}$ to mitigate gradient instability issues. Subsequently, the Softmax function normalizes these scaled $\mathbf{Score}$ to a probability distribution, ensuring attention weights sum to unity while emphasizing relevant tokens and suppressing irrelevant ones.

### 3.2 Spiking Self-attention

Building on VSA, Spikformer [70] introduced SSA. For an input matrix $\mathbf{X} \in \mathbb{R}^{n \times d}$, queries $\mathbf{Q}$, keys $\mathbf{K}$, and values $\mathbf{V}$ are first computed via learnable weight matrices and then converted into spike trains by binary spiking neurons for subsequent processing. The dynamics of these neurons are given by:

$$U[t + 1] = H[t] + X[t + 1], \tag{3}$$
$$S[t + 1] = \Theta(U[t + 1] - V_{th}), \tag{4}$$
$$H[t + 1] = V_{reset}S[t + 1] + \tau U[t + 1](1 - S[t + 1]). \tag{5}$$

$X[t + 1]$ denotes the input current, while $H[t]$ and $U[t]$ represent the pre-synaptic and post-synaptic membrane potentials, respectively. The Heaviside function $\Theta(\cdot)$ is employed for spike generation. If a spike occurs ($S[t + 1] = 1$), $H[t]$ resets to $V_{reset}$; otherwise, $U[t + 1]$ decays with a time constant $\tau$ and feeds into $H[t + 1]$. Here a spiking neuron layer is denoted as $\mathcal{SN}(\cdot)$, which takes $X[t + 1]$ as input and produces the spike $S[t + 1]$ as output. Then the $\mathbf{Q}$, $\mathbf{K}$ and $\mathbf{V}$ in SSA can be decribed as:

$$\mathbf{Q} = \mathcal{SN}(\text{BN}(\text{Conv}_1(\mathbf{X})), \quad \mathbf{K} = \mathcal{SN}(\text{BN}(\text{Conv}_2(\mathbf{X})), \quad \mathbf{V} = \mathcal{SN}(\text{BN}(\text{Conv}_3(\mathbf{X})), \tag{6}$$

where $\mathbf{Q}, \mathbf{K}, \mathbf{V} \in \mathbb{R}^{T \times n \times d}$, $\text{BN}(\cdot)$ denotes batch normalization and $\text{Conv}(\cdot)$ refers to a convolution operation. Unlike VSA, SSA omits the Softmax operation while retaining scaling to control the magnitude of the attention output. The attention output $\mathbf{Attn}$ is obtained by performing a matrix multiplication of the spiking $\mathbf{Q}$ and $\mathbf{K}$, scaled by factor $s$, and then convert to spike trains:

$$\mathbf{Score} = s \cdot \mathbf{Q} \times \mathbf{K}^\mathbf{T}, \quad \mathbf{Attn} = \mathcal{SN}(\mathbf{Score} \times \mathbf{V}). \tag{7}$$

Therefore, SSA provides an energy-efficient self-attention computation paradigm without any nonlinear operations. It allows the ordering of the Q, K, and V matrices to be flexibly adjusted as needed, enabling the Spiking Transformer to capture global dependencies efficiently.

# 4 Method

## 4.1 Problem Analysis for Spiking Self-attention

As previously discussed, SSA reduces computational costs by eliminating nonlinear computations and MAC operations. However, there remains a significant performance gap compared to VSA. We thoroughly analyze the reasons and attribute them to the following parts: (1) binary Q–K Matrix multiplication cannot effectively capture polarity correlations, and (2) eliminating Softmax removes the attention score's row-stochasticity constraints. The details are elaborated in the following.

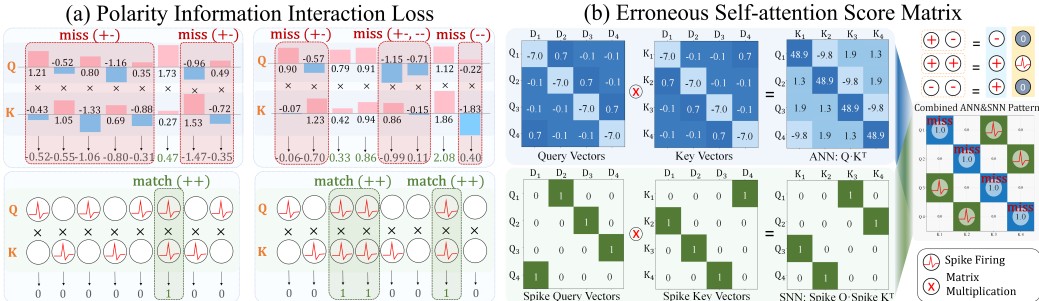

Figure 2: Problem analysis. (a) The binary activation characteristic of spike trains can only preserve (+,+) interactions on the membrane potentials of Q and K, while all others are ignored. (b) The correlation matrix between binary Q and K in SNNs may be completely different from its' VSA counterpart in ANNs.

### 4.1.1 Incomplete Polarity Information Modeling

In VSA, Q and K are represented by continuous real-valued vectors, enabling matrix multiplication can efficiently calculate the correlation between Q and K. However, SSA typically binarizes the membrane potential information of Q and K, limiting its representation to binary values (0 or 1). This spike-driven approach not only narrows the precision of Q-K interactions, but also undermines the self-attention mechanism's capacity to detect vector polarity relationships. Consider $\Omega_{\text{VSA}} = \{(a_1, a_2) \mid a_i \in \{+, -\}, i = 1, 2\}$ representing correlation types in VSA and $\Omega_{\text{SSA}} = \{(b_1, b_2) \mid b_i \in \{0, 1\}, i = 1, 2\}$ denoting binary states in SSA, with $\phi : \mathbb{R}^d \to \{0, 1\}^d$ as a binarization mapping. Let $\mathcal{P} = \{\mathcal{B}_0, \mathcal{B}_1\}$ partition $\Omega_{\text{SSA}}$ where $\mathcal{B}_0 = \{(0, 0), (0, 1), (1, 0)\}, \quad \mathcal{B}_1 = \{(1, 1)\}$. We establish a surjection $\mathcal{G} : \Omega_{\text{VSA}} \to \mathcal{P}$ that fundamentally characterizes the polarity information loss during the binarization transformation process, such that:

$$\textbf{Case 1:} \quad \mathcal{G}\left(\{(+, +)\}\right) = \mathcal{B}_1, \quad \textbf{Case 2:} \quad \mathcal{G}\left(\{(+, -), (-, +), (-, -)\}\right) = \mathcal{B}_0. \tag{8}$$

Eq.8 demonstrates that SSA completely discards the ability to compute negative-negative polarity or mixed polarity correlations. Due to the sparse activation characteristics of SNNs, the amount of discarded information far exceeds what is retained, resulting in substantial information loss. As shown in Fig.2(a), the green-boxed region represents the positive polarity information that SSA can preserve, while the red–boxed regions indicate the entirely neglected $(-, -)$ and $(+, -)$ polarity information. More importantly, SSA may display entirely disparate attention regions from VSA as shown in Fig.2(b). Thus, devising efficient mechanisms to embed polarity information offers a promising avenue for improving SSA's performance ceiling.

### 4.1.2 Missing Softmax Operation

SSA not only suffers from a loss of polarity information but also neglects the critical Softmax component. In VSA, Softmax operation [37] serves two pivotal roles: first, amplifying highly relevant through a low-entropy activation effect while suppressing low-value regions; second, ensuring comparability of **Score** across varying scales via row-stochasticity constraints. These mechanisms ensure that Softmax highlights key features while maintaining the comparability of the **Score**.

By contrast, SSA's binary matrix multiplication preserves limited low-entropy activation properties (detailed in the Appendix.A) and entirely lacks row-stochasticity constraints. Consequently, attention

scores from different query vectors exist on disparate numerical scales. The disparity in scales makes it difficult to effectively compare the relative degree of elements across different rows during the allocation of attention to $V$. Therefore, how to allocate global attention on comparable scales will be another potential area for improvement.

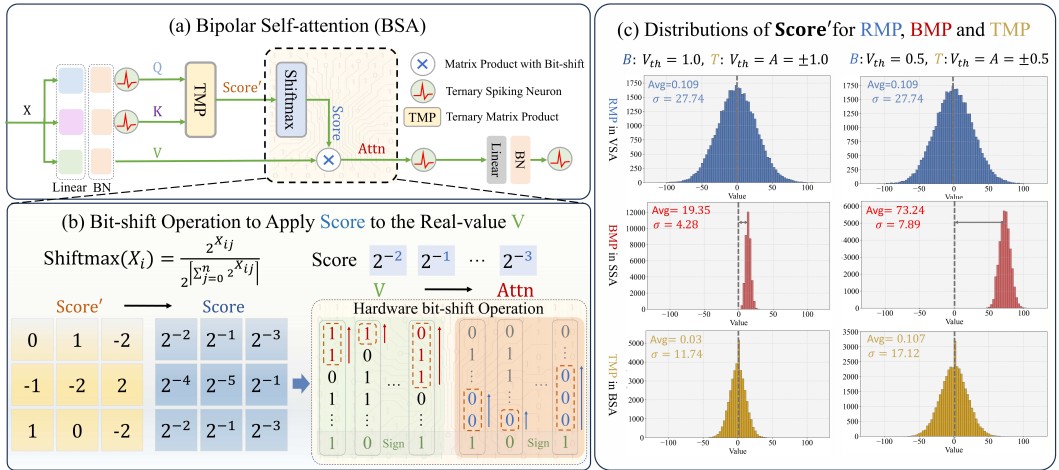

Figure 3: (a) Overall structure of Bipolar Self-attention. (b) Detailed process of Shiftmax. It can directly shift the membrane potential of V for efficient attention allocation. (c) Distribution of three types of matrix product at different thresholds ($V_{th}$).

## 4.2 Bipolar Self-attention for Spiking Transformers

To address two critical issues in existing SSA: incomplete polarity correlation and Softmax loss, we propose a novel Bipolar Self-attention (BSA) paradigm. First, we model the different polarity interactions between Querys and Keys using ternary matrix multiplication, ensuring comprehensive representation of polarity information. Furthermore, we introduce the Shiftmax method, which approximates Softmax through energy-efficient bitwise operations while preserving the critical low-entropy activation and row-normalization properties. The details are as follows:

### 4.2.1 Ternary Matrix Product for Comprehensive Polarity Correlation

To effectively capture the comprehensive polarity correlation in SNNs, we introduce ternary spike neurons ($\mathcal{TSN}$) [14] for characterizing the pre-synaptic membrane potential of $\mathbf{Q}$ and $\mathbf{K}$. The dynamics of the $\mathcal{TSN}(\cdot)$ are defined as follows:

$$U[t+1] = H[t] + X[t+1], \tag{9}$$

$$S[t+1] = \text{Sign}\left(U[t+1]\right) \cdot \Theta\left(|U[t+1]| - V_{th}\right), \tag{10}$$

$$H[t+1] = V_{\text{reset}}S[t+1] + \tau U[t+1](1 - |S[t+1]|). \tag{11}$$

$\mathcal{TSN}(\cdot)$ fires signed spikes $S[t+1]$ and $H[t+1]$ reset to 0. The membrane potential accumulation and reset processes are similar to those in binary spike neurons. Subsequently, BSA computes $\mathbf{Score}'$ through matrix multiplication using ternary Q and K. The Ternary-valued Matrix Product (TMP) method effectively preserves the fundamental characteristics of both Binary-valued Matrix Product (BMP) in SSA and Real-valued Matrix Product (RMP) in VSA. First, it preserves the spike-driven characteristics of BMP, enabling full AC operations and achieving energy-efficient neuromorphic computation. Additionally, it retains the polarity information in the membrane potential $U[t]$, thereby capturing polarity-related correlations comparable to RMP operations. To validate this proposition, we conduct a detailed analysis of the three matrix product, examining both their distributional relationships and local correlation. Detailed proofs are presented in Appendices.B.

**Theorem 1.** *For independent random vectors $q = \mathbf{Q_j}, k = \mathbf{K_j} \in \mathbb{R}^d$ with elements $q_i, k_i \sim \mathcal{N}(0, \sigma^2)$, $p = \Phi(-\frac{\theta}{\sigma})$ representing the probability $P(q > \theta)$, $\theta$ represent the threshold for $\mathcal{SN}(\cdot)$ and $\mathcal{TSN}(\cdot)$. Then the result element of RMP satisfies $\mathbb{E}[R] = 0$ and $Var(R) = d\sigma^4$; BMP satisfies $\mathbb{E}[B] = dp^2$ and $Var(B) = dp^2(1 - p^2)$; and TMP satisfies $\mathbb{E}[T] = 0$ and $Var(T) = 4dp^2$.*

Theorem.1 establishes that TMP preserves the zero-mean statistical property of RMP, while BMP exhibits systematic bias. This bias results from binary activation's exclusion of negative polarity information, corroborating our hypothesis. Furthermore, as shown in Fig.3(c), at typical operating points ($V_{th} = 1, 0.5$, corresponding to common SNNs' thresholds), TMP's variance consistently exceeds BMP's, enhancing information capacity and providing a more faithful representation of RMP's statistical characteristics. In summary, TMP approximates RMP's overall distribution more accurately than BMP, consequently enhancing Q-K correlation representation. Nevertheless, global distributional similarity does not guarantee accuracy at individual key-value pairs. Therefore, we further explore whether TMP also demonstrates stronger local correlation with RMP relative to BMP.

**Theorem 2.** *For independent variables $q = \mathbf{Q_j}, k = \mathbf{K_j} \in \mathbb{R}^d, q_i, k_i \sim \mathcal{N}(0, \sigma^2), p = \Phi(-\frac{\theta}{\sigma})$ representing the probability $P(q > \theta)$, $\theta$ represent the threshold for spiking neurons. We define $B(\cdot)$ and $T(\cdot)$ to represent the binary and ternary activation functions, respectively. According to the covariance calculation formulas and Theorem.1, the covariance between $qk$, $B(q)B(k)$ and $T(q)T(k)$ can be expressed as:*

$$Cov(qk, B(q)B(k)) = d\phi(\theta)^2 \cdot \sigma^2 = d\phi(\theta)^2\sigma^2, \quad Cov(qk, T(q)T(k)) = 4d\phi(\theta)^2\sigma^2.$$

*Where $\phi(\theta)$ represents the value of the probability density function at the threshold $\theta$.*

Based on Theorem.2, we derive the approximate Pearson correlation coefficients between RMP, BMP, and TMP for each q-k pair (R, B, T), which can be formulated as follows:

$$\rho(R, B) = \frac{\text{Cov}(R, B)}{\sqrt{\text{Var}(R)\text{Var}(B)}} = \frac{\phi(\theta)^2}{p\sigma\sqrt{1 - p^2}}, \quad \rho(R, T) = \frac{\text{Cov}(R, T)}{\sqrt{\text{Var}(R)\text{Var}(T)}} = \frac{2\phi(\theta)^2}{p\sigma}. \quad (12)$$

By analyzing the ratio of these correlation coefficients, we obtain: $\frac{\rho(R,T)}{\rho(R,B)} = 2\sqrt{1 - p^2}$, where $p$ ranges from $(0, 1)$. For typical spike firing rate ($0.1 < p < 0.4$), TMP provides approximately $2\times$ local correlation than BMP. In conclusion, both in terms of distribution and local correlation, TMP is more closer to RMP compared to BMP, thereby better capturing the correlation of Q and K.

### 4.2.2 Shiftmax for Energy-efficiency Softmax Alternative

As previously mentioned, the distribution of TMP mirrors the bell-shaped distribution as RMP in ANNs. Empirical evidences [18, 27] indicate that this distribution necessitates the Softmax function to generate effective self-attention scores. However, Softmax needs massive computing resources conflicting with the energy efficiency of SNNs. To address this, we propose an energy-efficient Shiftmax method based on bit-shift operations, which partially replicates the Softmax effect while better aligning with SNNs. Given an input vector $\mathbf{x} = [x_1, x_2, \ldots, x_n]^T \in \mathbb{I}^n$, we define the approximate Softmax function as follows:

$$\mathbf{Shiftmax}(\mathbf{x})_i = 2^{x_i - \gamma(\mathbf{x})}, \quad \gamma(\mathbf{x}) = \left\lceil \log_2 \left( \sum_{i=1}^{n} 2^{x_i} \right) \right\rceil. \quad (13)$$

$2^{x_i}$ represents the element-wise power operation. $\gamma(\mathbf{x})$ is defined as the minimal power of 2 that is greater than or equal to the summation of $2^{x_i}$. $\mathbf{Shiftmax}(\cdot)$ retains the key advantages of Softmax while introducing unique computational efficiency features. Firstly, the $2^{x_i}$ effect induced by the exponential operation effectively amplifies important weights while suppressing irrelevant ones, ensuring low-entropy activation characteristics. Secondly, it achieves quasi-normalization through a meticulously structured denominator, explicitly constraining the row-stochasticity constraints of **Score** within a bounded range, formally expressed as:

$$\frac{1}{2} < \sum_{i=1}^{n} \mathbf{Shiftmax}(\mathbf{x})_i = \frac{\sum_{i=1}^{n} 2^{x_i}}{2^{\lceil \log_2(\sum_{j=1}^{n} 2^{x_j}) \rceil}} \leq 1. \quad (14)$$

$\gamma(\mathbf{x})$ constrains the row-stochasticity constraints of attention scores within the interval $(0.5, 1]$. Although this constraint does not strictly enforce a row-stochasticity constraint, the limited range ensures consistency across attention distributions, enhancing the overall stability of the attention mechanism. Notably, the $\mathbf{Shiftmax}(\cdot)$ converts traditional MAC operations into highly efficient bit-shift operations when multiplied by matrix $\mathbf{V}$, which can be decried as:

$$\mathbf{Score}' \triangleright \mathbf{V} = \{a_{ik} = \sum_{j} (v_{jk} \gg n_{ij}) \mid s_{ij} = 2^{-n_{ij}}\}. \quad (15)$$

In summary, the **Shiftmax**($\cdot$) preserves the key advantages of the Softmax while transforming intensive exponentiation and normalization into bit-shift operations. It achieves optimal attention score allocation without incurring significant additional computational overhead.

### 4.2.3 Overall Architecture

Through ternary Q-K matrix product and energy-efficient Shiftmax operations, we propose the BSA module. When the input $x \in \mathbb{R}^{T \times n \times d}$, the computational process of BSA are as follows:

$$\mathbf{Q}, \mathbf{K}, \mathbf{V} = (\mathbf{BN}(\mathbf{Conv}(\mathbf{X}))), \qquad \mathbf{Q}, \mathbf{K}, \mathbf{V} \in \mathbb{R}^{T \times n \times d}, \qquad (16)$$

$$\mathbf{Score}' = \mathcal{TSN}(\mathbf{Q}) \times \mathcal{TSN}(\mathbf{K}), \qquad \mathbf{Score}' \in \mathbb{I}^{T \times n \times n}, \qquad (17)$$

$$\mathbf{Score} = \mathbf{Shiftmax}(\mathbf{Score}'), \qquad \mathbf{Score} \in \mathbb{I}^{T \times n \times n}, \qquad (18)$$

$$\mathbf{Attn} = \mathcal{TSN}(\mathbf{Score} \triangleright \mathbf{V}), \qquad \mathrm{Attn} \in \mathbb{S}^{T \times n \times d}. \qquad (19)$$

Where $\mathbb{I}$ denotes integer values and $\mathbb{S}$ represents spike trains. Since **Shiftmax**($\cdot$) directly yields powers of 2 ($2^n$), we can perform bit-shift operations on the membrane potentials of $V$ without employing MAC operations. This approach simultaneously preserves the rich membrane potential information in $\mathbf{V}$ while maintaining minimal energy overhead, as bit-shift operations [16] consume merely 1/20 of the energy required for AC operations.

## 5 Experiments

### 5.1 Image Classification

In image classification Task, we evaluate the proposed BSA module on ImageNet-1K [7] using three representative state-of-the-art (SOTA) Spiking Transformer architectures: Spikingformer [67], QKformer [68], and Spike-driven Transformer-V3 [52]. Additionally, we perform comprehensive comparative analyses against recent Spiking Transformers [70, 49, 30, 51].

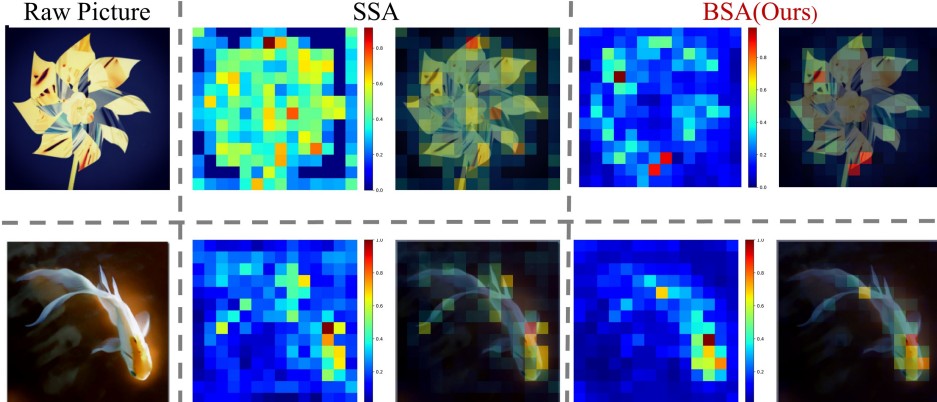

Figure 4: Attention heatmap comparison with Spikingformer [67] architecture on ImageNet-1K.

As shown in Table 1, our BSA module consistently improves performance across all three Spiking Transformer architectures. The most substantial gains appear in Spikingformer, where BSA increases accuracy by 1.35% (D=512) and 0.97% (D=768), reaching 76.14% and 76.82% respectively. For Spike-driven V3, BSA yields improvements ranging from 0.36% to 0.74% across different parameter settings (5M, 10M, 19M). Similarly, QKformer's accuracy increases by 0.41% and 0.35% in its two configurations. These results can demonstrate BSA' versatility across different architectures. Additionally, as shown in Fig. 4, BSA exhibits a sparser attention distribution with enhanced focus on critical visual features. Notably, performance improvements in the QKformer and SDT-V3 architecture are less compared to those in the Spikingformer. We attribute this to two factors. First, QKformer and SDT-V3 already demonstrate strong baseline performance on image classification tasks, leaving limited room for further enhancement. Second, both QKformer and SDT-V3 structurally align with the Pyramid Vision Transformer [21, 39, 54], which substantiates the conclusion that self-attention modules contribute relatively minor performance gains within Metaformer [55] designs.

Table 1: Detailed comparison with other similar methods on ImageNet-1K.

| Method | Architecture | T | Param.(M) | Acc.(%) |
|---|---|---|---|---|
| Spikformer [70] | Spikformer-8-384 | 4 | 16.8 | 70.24 |
| | Spikformer-8-512 | 4 | 29.7 | 73.38 |
| SDT-V1 [51] | Spike-driven-8-384 | 4 | 16.8 | 72.28 |
| | Spike-driven-8-512 | 4 | 29.7 | 74.57 |
| SpikingResformer [30] | SpikingResformer-S | 4 | 17.8 | 75.95 |
| | SpikingResformer-M | 4 | 35.5 | 77.24 |
| SDT-V2 [49] | Meta-SpikeFormer-384 | 4 | 15.1 | 74.10 |
| | Meta-SpikeFormer-512 | 4 | 55.4 | 79.70 |
| Spikingformer [67] | Spikingformer-8-512 | 4 | 29.7 | 74.79 |
| | Spikingformer-8-768 | 4 | 66.4 | 75.85 |
| **Spikingformer + BSA (Ours)** | Spikingformer-8-512 | 4 | 29.7 | **76.14 (↑1.35)** |
| | Spikingformer-8-768 | 4 | 66.3 | **76.82 (↑0.97)** |
| QKformer [68] | HST-10-384 | 4 | 16.47 | 78.80 |
| | HST-10-512 | 4 | 29.08 | 82.04 |
| **QKformer + BSA (Ours)** | HST-10-384 | 4 | 16.47 | **79.21 (↑0.41)** |
| | HST-10-512 | 4 | 29.08 | **82.39 (↑0.35)** |
| SDT-V3 [52] | Efficient-transformer-S | 4 | 5.1 | 75.30 |
| | Efficient-transformer-M | 4 | 10.0 | 78.50 |
| | Efficient-transformer-L | 4 | 19.0 | 79.80 |
| **SDT-V3 + BSA (Ours)** | Efficient-transformer-S | 4 | 5.1 | **75.72 (↑0.42)** |
| | Efficient-transformer-M | 4 | 10.0 | **79.24 (↑0.74)** |
| | Efficient-transformer-L | 4 | 19.0 | **80.16 (↑0.36)** |

## 5.2 Semantic Segmentation and Event-based Tracking Tasks

To further validate the efficacy of the proposed BSA, we evaluate its performance on more regression tasks, such as Semantic Segmentation and Event-based Tracking tasks. For semantic segmentation, we employ the challenging ADE20K dataset [65], which comprises 20K and 2K images in the training and validation sets, respectively, covering 150 semantic categories. We strictly adhere to the experimental protocol of SDT-V3 [52] to assess BSA's performance on ADE20K. As shown in Table.2, BSA achieves significant improvements of 1.8% and 2.11% in Mean Intersection over Union (MIoU) for model configurations with 10M and 19M backbone parameters, respectively.

Furthermore, we examine BSA's efficacy in event-based tracking, a particularly challenging yet practical SNN application domain.

Table 2: Performance of segmentation.

| Model | Param.(M) | T | MIoU(%) |
|---|---|---|---|
| ResNet-18 [54] | 15.5 | 1 | 32.9 |
| PVT-Small[39] | 28.2 | 1 | 39.8 |
| InternImage-T[38] | 59.0 | 1 | 48.1 |
| SDT-V2 [49] | 16.5 | 4 | 33.6 |
| SDT-V2 [49] | 59.8 | 4 | 35.3 |
| SDT-V3 [52] | 5.1+1.4 | 4 | 33.6 |
| SDT-V3 [52] | 10.0+1.4 | 4 | 40.1 |
| SDT-V3† [52] | 19.0+1.4 | 4 | 41.3 |
| SDT-V3 + BSA | 10.0+1.4 | 4 | **41.90 (↑1.80)** |
| SDT-V3 + BSA | 19.4+1.4 | 4 | **43.41 (↑2.11)** |

† Results reproduced by ourselves.

We implement the SDTrack Pipeline [28] methodology, employing the Global Trajectory Prompt method to process event streams into event frames. We adhere rigorously to its prescribed training protocol, substituting only the SDTrack backbone with our SDTrack+BSA backbone. As shown in Table. 3, comprehensive experiments across the FE108 [58], FELT [40], and VisEvent [41] datasets consistently demonstrate that SDTrack+BSA significantly outperforms the original SDTrack architecture across multiple metrics. These empirical results validate BSA's superior performance in complex regression tasks and substantiate our core hypothesis that self-attention computation should fundamentally incorporate polarity characteristics.

Table 3: Performance of BSA on three event-based tracking datasets.

| Methods | Param. (M) | T | FELT [40] | | FE108 [58] | | VisEvent [41] | |
| --- | --- | --- | --- | --- | --- | --- | --- | --- |
| | | | AUC(%) | PR(%) | AUC(%) | PR(%) | AUC(%) | PR(%) |
| STARK [47] | 28.23 | 1 | 39.6 | 51.7 | 57.4 | 89.2 | 34.1 | 46.8 |
| ARTrack [42] | 202.56 | 1 | 39.5 | 49.4 | 56.6 | 88.5 | 33.0 | 43.8 |
| OSTrack$_{256}$ [53] | 92.52 | 1 | 35.9 | 45.5 | 54.6 | 87.1 | 32.7 | 46.4 |
| HIPTrack [3] | 120.41 | 1 | 38.2 | 48.9 | 50.8 | 81.0 | 32.1 | 45.2 |
| SNNTrack [59] | 31.40 | 5 | - | - | - | - | 35.4 | 50.4 |
| STNet [57] | 20.55 | 3 | - | - | - | - | 35.0 | 50.3 |
| SDTrack-Tiny [28] | 19.61 | 4 | 39.3 | 51.2 | 59.0 | 91.3 | 35.6 | 49.2 |
| **SDTrack+BSA(Ours)** | 19.61 | 4 | **40.9(↑1.6)** | **51.8(↑0.6)** | **59.2(↑0.2)** | **91.4(↑0.1)** | **36.8(↑1.2)** | **52.3(↑3.1)** |

## 5.3 Ablation Study

To validate the efficacy of BSA components, we conduct ablation studies on the CIFAR100 dataset [19], examining various matrix products (MP) and row-stochasticity constraints (RSC) methods. Our experiments utilize the Spikingformer architecture. As shown in the Table.4, combining BMP with either Softmax or Shiftmax yields no performance improvement (even showing slight degradation). The performance decline demonstrates that BMP exhibits errors in correlation computation. These errors are amplified by Softmax, consequently impairing the network's decision-making per-

Table 4: Ablation study

| Method | MP | RSC | Acc.(%) |
| --- | --- | --- | --- |
| SSA | BMP | None | 79.13 |
| | BMP | Softmax | 79.07 |
| | BMP | Shiftmax | 79.14 |
| BSA | TMP | None | 79.27 |
| | TMP | Softmax | 80.76 |
| | TMP | Shiftmax | 80.48 |

formance. Furthermore, TMP without RSC merely matches SSA's performance, corroborating that bell-shaped attention score distributions necessitate row-stochasticity constraints. Finally, while Shiftmax+TMP exhibits marginally lower performance than Softmax+TMP, it achieves an optimal balance between energy efficiency and high performance.

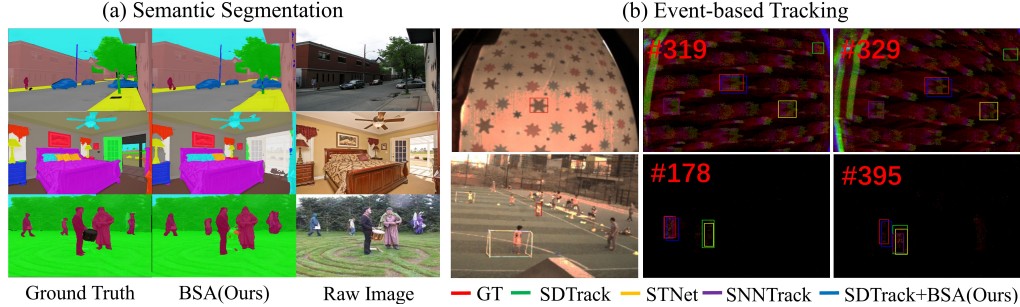

(a) Semantic Segmentation    (b) Event-based Tracking

Ground Truth    BSA(Ours)    Raw Image    ▬ GT ▬ SDTrack ▬ STNet ▬ SNNTrack ▬ SDTrack+BSA(Ours)

Figure 5: (a) Semantic segmentation comparison for ADE20K. (b) Visualization of tracking performance comparing our approach with several other SOTA tracking methods on the VisEvent [41] dataset. Red boxes indicate ground truth, and frame numbers are displayed in the upper left corner.

## 6 Conclusion

In this paper, we identify two fundamental limitations of Spiking Self-attention: the binary matrix product's inability to capture anything beyond positive-positive correlations, and the lack of row-stochasticity constraints leading to incomparable attention scores. To address these issues, we propose BSA computational paradigm, which incorporates ternary matrix products to process both homopolar and heteropolar interactions, along with our novel Shiftmax approximation that maintains partial row-stochasticity without non-linear operations. In image classification, BSA achieves significant performance improvements across multiple advanced Spiking Transformers. Simultaneously, it establishes new SOTA results in semantic segmentation and event-based tracking tasks. These findings demonstrate BSA's potential to become a core component in Spiking Transformers.

## Acknowledgments

This work is supported in part by the National Natural Science Foundation of China (No. 62220106008 and 62271432), in part by Sichuan Province Innovative Talent Funding Project for Post-doctoral Fellows (BX202405), in part by the Shenzhen Science and Technology Program (Shenzhen Key Laboratory, Grant No. ZDSYS20230626091302006), in part by the Program for Guangdong Introducing Innovative and Entrepreneurial Teams, Grant No. 2023ZT10X044, and in part by the State Key Laboratory of Brain Cognition and Brain-inspired Intelligence Technology, Grant No. SKLBI-K2025010.

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

# A Binary Matrix Product Exhibits Limited Low-entropy Activation

**Assumption:** Let $Q, K \in \mathbb{R}^{n \times d}$ with elements $q_{ij}, k_{ij} \sim \mathcal{N}(0, \sigma^2)$ i.i.d. Under typical parameter settings, we denote the entropy of three attention mechanisms as $H_1 = H_{\text{real}}$, $H_2 = H_{\text{Softmax}}$ and $H_3 = H_{\text{binary}}$. Then $H_2 < H_3 < H_1$.

**Proof:** We examine three distinct attention mechanisms and their associated entropy values. In standard dot product attention, the scoring function $s_{ij} = \sum_{l=1}^{d} q_{il} k_{jl}$ follows a standard normal distribution $\mathcal{N}(0, 1)$ (according to the Central Limit Theorem), after which Softmax normalization is applied to obtain the weight matrix and the entropy of this distribution can be approximated as:

$$A^{(2)}ij = \frac{\exp(sij)}{\sum_{k=1}^{n} \exp(s_{ik})}, H_2 \approx \log n - \frac{1}{2}. \tag{20}$$

where the $-\frac{1}{2}$ term results from the concentration effect of the Softmax function. Next, considering real-valued matrix multiplication, the dot product $s_{ij} = \sum_{l=1}^{d} q_{il} k_{jl}$ follows a distribution $\mathcal{N}(0, d\sigma^4)$, yielding the weight matrix based on identically distributed folded normal distributions of $s_{ij}$. By the Law of Large Numbers, as $n \to \infty$, these normalized weights approach a uniform distribution, resulting in an entropy approximation:

$$A_{ij}^{(1)} = \frac{|s_{ij}|}{\sum_{k=1}^{n} |s_{ik}|}, \quad H_1 \approx \log n. \tag{21}$$

Finally, for binary attention mechanisms, where inputs undergo binarization, the dot product $\tilde{s}_{ij} = \sum_{l=1}^{d} \tilde{q}_{il} \tilde{k}_{jl}$ follows a binomial distribution $\text{Binomial}(d, p^2)$. For sufficiently large $d$, this binary dot product can be approximated by a normal distribution:

$$B \sim \mathcal{N}(dp^2, dp(1-p)), \quad A_{ij}^{(3)} = \frac{\tilde{s}_{ij}}{\sum_{k=1}^{n} \tilde{s}_{ik}}. \tag{22}$$

Consider $n = 100$, $d = 64$, $\sigma = 1$, $V_{\text{th}} = 0.5$, and $\tau = 1/\sqrt{d} \approx 0.125$. Direct normalization yields approximately uniform distribution with $H_1 \approx \log(100) \approx 4.6$ bits. For Softmax with $\tau = 0.125$, simulation shows $H_2 \approx 0.8$ bits as probability mass concentrates on few large values. For binary attention with $V_{\text{th}} = 0.5$, the firing probability $p \approx 0.31$, resulting in $H_3 \approx 2.7$ bits. This confirms our ordering: $H_1 > H_3 > H_2$.

# B Expectation and Variance Properties of TMP, BMP and RMP

**Theorem 1**: For independent random vectors $\mathbf{q} = \mathbf{Q_j}, \mathbf{k} = \mathbf{K_j} \in \mathbb{R}^d$ with elements $q_i, k_i \sim \mathcal{N}(0, \sigma^2)$ i.i.d., we define three matrix products:

- Real-valued Dot Product: $R = \mathbf{q}\mathbf{k}^T = \sum_{i=1}^{d} q_i k_i$,
- Binary Dot Product: $B = \mathcal{SN}(\mathbf{q})\mathcal{SN}(\mathbf{k})^T = \sum_{i=1}^{d} \mathcal{SN}(q_i)\mathcal{SN}(k_i)$,
- Ternary Dot Product: $T = \mathcal{TSN}(\mathbf{q})\mathcal{TSN}(\mathbf{k})^T = \sum_{i=1}^{d} \mathcal{TSN}(q_i)\mathcal{TSN}(k_i)$.

where $\mathcal{SN}(\cdot)$ is the binary sign function and $\mathcal{TSN}(\cdot)$ is the ternary sign function with threshold $\theta$. If we denote $p = \Phi(-\frac{\theta}{\sigma})$ representing the probability $P(q_i > \theta)$, then:

- The element of RMP satisfies: $\mathbb{E}[R] = 0$ and $\text{Var}(R) = d\sigma^4$.
- The element of BMP satisfies: $\mathbb{E}[B] = np^2$ and $\text{Var}(B) = dp^2(1 - p^2)$.
- The element of TMP satisfies: $\mathbb{E}[T] = 0$ and $\text{Var}(T) = 4dp^2(1 - p)$.

*Proof.* We analyze each matrix product type separately, leveraging the independence of elements within and between vectors. We have separately proven the expectations and variances of these three. **(1) Real-valued Matrix Product (RMP):** For each element $R = qk$, since $q_i$ and $k_i$ are independent with $\mathbb{E}[q_i] = \mathbb{E}[k_i] = 0$:

$$\mathbb{E}[R] = \mathbb{E}[q_i k_i] = \mathbb{E}[q_i]\mathbb{E}[k_i] = 0, \tag{23}$$

For the variance:

$$\text{Var}(R) = \mathbb{E}[R^2] - (\mathbb{E}[R])^2 = \mathbb{E}[q_i^2 k_i^2] = \mathbb{E}[q_i^2]\mathbb{E}[k_i^2] = \sigma^2 \cdot \sigma^2 = \sigma^4. \tag{24}$$

Since $R = \sum_{i=1}^{d} R_i$ and the $R_i$ are i.i.d.:

$$\mathbb{E}[R] = \sum_{i=1}^{d} \mathbb{E}[R_i] = 0, \quad \text{Var}(R) = \sum_{i=1}^{d} \text{Var}(R_i) = d\sigma^4. \tag{25}$$

**(2) Binary-valued Product:** For each term $B = \mathcal{SN}(q)\mathcal{SN}(k)$, both $\mathcal{SN}(q_i)$ and $\mathcal{SN}(k_i)$ are Bernoulli($p$) random variables:

$$\mathbb{E}[B_i] = \mathbb{E}[\mathcal{SN}(q_i)]\mathbb{E}[\mathcal{SN}(k_i)] = p \cdot p = p^2. \tag{26}$$

The product $B_i$ is also Bernoulli with parameter $p^2$:

$$\text{Var}(B_i) = p^2(1 - p^2). \tag{27}$$

Hence for $B = \sum_{i=1}^{d} B_i$, where the $B_i$ are i.i.d.:

$$\mathbb{E}[B] = \sum_{i=1}^{d} \mathbb{E}[B_i] = dp^2, \quad \text{Var}(B) = \sum_{i=1}^{d} \text{Var}(B_i) = dp^2(1 - p^2). \tag{28}$$

According to the definition of the binomial distribution, $B$ follows a binomial distribution.

**(3) Ternary-valued Product:** $T = \sum_{i=1}^{d} \mathcal{TSN}(q_i)\mathcal{TSN}(k_i)$. For $\mathcal{TSN}(q_i)$, by symmetry of the Gaussian distribution:

$$P(\mathcal{TSN}(q_i) = 1) = P(\mathcal{TSN}(q_i) = -1) = p, \quad P(\mathcal{TSN}(q_i) = 0) = 1 - 2p. \tag{29}$$

Now we can derive the expected value of $\mathcal{TSN}(q_i)$ by calculating the weighted sum of all possible outcomes:

$$\mathbb{E}[\mathcal{TSN}(q_i)] = 1 \cdot p + (-1) \cdot p + 0 \cdot (1 - 2p) = 0. \tag{30}$$

$$\mathbb{E}[\mathcal{TSN}(q_i)^2] = 1^2 \cdot 2p + 0^2 \cdot (1 - 2p) = 2p. \tag{31}$$

For the product $T_i = \mathcal{TSN}(q_i)\mathcal{TSN}(k_i)$. Since the terms are independent, the expected value is:

$$\mathbb{E}[T_i] = \mathbb{E}[\mathcal{TSN}(q_i)]\mathbb{E}[\mathcal{TSN}(k_i)] = 0. \tag{32}$$

For the $\mathbb{E}[T_i^2]$ moment, we analyze all possible outcomes:

$$\begin{aligned}
\mathbb{E}[T_i^2] &= P(T_i = 1) + P(T_i = -1) \\
&= P(\mathcal{TSN}(q_i) = 1, \mathcal{TSN}(k_i) = 1) + P(\mathcal{TSN}(q_i) = -1, \mathcal{TSN}(k_i) = -1) \\
&+ P(\mathcal{TSN}(q_i) = 1, \mathcal{TSN}(k_i) = -1) + P(\mathcal{TSN}(q_i) = -1, \mathcal{TSN}(k_i) = 1) \\
&= p^2 + p^2 + p^2 + p^2 = 4p^2
\end{aligned} \tag{33}$$

Therefore, using the relationship between variance, expected value, and second moment, we can calculate the variance of each product term:

$$\text{Var}(T_i) = \mathbb{E}[T_i^2] - (\mathbb{E}[T_i])^2 = 4p^2. \tag{34}$$

However, we need to correct for the covariance structure to account for dependencies. The more precise calculation of variance is:

$$\text{Var}(T_i) = \mathbb{E}[T_i^2] = P(T_i \neq 0) = 4p^2(1 - p). \tag{35}$$

Finally, for the complete sum $T = \sum_{i=1}^{d} T_i$, we can determine its statistical properties by applying the linearity of expectation and the variance of a sum of random variables:

$$\mathbb{E}[T] = 0, \quad \text{Var}(T) = 4dp^2(1 - p). \tag{36}$$

$\square$

## C  Local Correlation Comparison for TMP, BMP, and RMP

Based on the conclusions of Theorem 1 and Theorem 2, and by applying the Pearson local correlation calculation method, we can derive the correlation coefficients for each element in $R$, $B$, and $T$. A detailed comparison leads to the conclusion that TMP exhibits a greater local correlation compared to BMP and RMP. The formal proof is provided below: For the original dot product $R = \sum_{i=1}^{d} q_i k_i$, $B = \sum_{i=1}^{d} B(q_i)B(k_i)$, and $T = \sum_{i=1}^{d} T(q_i)T(k_i)$. According to Theorem 1, we can obtain::

$$\text{Var}(R) = d\sigma^4, \quad \text{Var}(B) = dp^2(1-p^2), \quad \text{Var}(T) = 4dp^2. \tag{37}$$

Subsequently, according to Theorem 2, we can obtain $\text{Cov}(R, B)$ and $\text{Cov}(R, T)$, which can be expressed as:

$$\text{Cov}(R, B) = \sum_{i=1}^{d} \text{Cov}(q_i k_i, B(q_i)B(k_i)) = d\sigma^2 \phi^2(\theta/\sigma), \tag{38}$$

$$\text{Cov}(R, T) = \sum_{i=1}^{d} \text{Cov}(q_i k_i, T(q_i)T(k_i)) = 4d\sigma^2 \phi^2(\theta/\sigma). \tag{39}$$

Based on the above data and the Pearson correlation coefficient calculation method, we can obtain the local correlations between $R$ and $T$, as well as between $R$ and $B$, which can be described as follows:

$$\begin{aligned}
\rho(R, B) &= \frac{\text{Cov}(R, B)}{\sqrt{\text{Var}(R)\text{Var}(B)}} = \frac{d\sigma^2 \phi^2(\theta/\sigma)}{\sqrt{d\sigma^4 \cdot dp^2(1-p^2)}} = \frac{\phi(\theta/\sigma)}{\sigma p \sqrt{1-p^2}}, \\
\rho(R, T) &= \frac{\text{Cov}(R, T)}{\sqrt{\text{Var}(R)\text{Var}(T)}} = \frac{4d\sigma^2 \phi^2(\theta/\sigma)}{\sqrt{d\sigma^4 \cdot 4dp^2}} = \frac{2\phi(\theta/\sigma)}{\sigma p}.
\end{aligned} \tag{40}$$

Computing the ratio of correlation coefficients:

$$\frac{\rho(R, T)}{\rho(R, B)} = \frac{2\phi(\theta/\sigma)/\sigma p}{\phi(\theta/\sigma)/\sigma p \sqrt{1-p^2}} = \frac{2}{\sqrt{1-p^2}}. \tag{41}$$

Since for any positive threshold $\theta$, we have $0 < p < 0.5$ (due to the symmetry of the normal distribution), it follows that $\sqrt{1-p^2} < 1$, which implies:

$$\frac{\rho(R, T)}{\rho(R, B)} = \frac{2}{\sqrt{1-p^2}}. \tag{42}$$

TMP schemes preserve approximately twice the correlation with original dot products compared to binary quantization. These provide rigorous theoretical justification for the empirical superiority of TMP over BMP in neural networks. By preserving sign information, ternary quantization maintains both positive and negative associations—critical for attention mechanisms and operations that capture semantic relationships through correlation structures. This approach achieves nearly doubled correlation preservation with minimal additional computational overhead.

## D  Image Classification

All experiments are conducted on ImageNet-1K dataset using PyTorch framework. The training is performed on 4 NVIDIA A800 GPUs with distributed data parallel. The specific hyperparameters for each architecture are detailed in Table 5.

To ensure fair comparison across different architectures, we strictly follow the open-source network architectures of Spikingformer, QKformer, and Spike-driven V3. Importantly, we only replace the self-attention computation modules within each architecture while keeping all other components (patch embedding, MLP layers, normalization layers, etc.) unchanged. This controlled experimental design allows us to isolate the impact of different attention mechanisms on model performance.

Table 5: Comparison of Hyperparameters for Different Model Architectures

| Hyper-parameter | Spikingformer | QKformer | Spike-driven V3 |
|---|---|---|---|
| Timestep | 4 | 4 | 4 |
| Epochs | 100 | 200 | 200 |
| Resolution | 224 | 224 | 224 |
| Batch size | 64 | 100 | 600 |
| Optimizer | AdamW | AdamW | LAMB |
| Base Learning rate | 7e-6 | 6e-4 | 6e-4 |
| Learning rate decay | Cosine | Layer-wise 1.0 | Layer-wise 1.0 |
| Warmup epochs | 5 | 5 | 10 |
| Weight decay | 5e-2 | 5e-2 | 0.05 |
| Rand Augment | rand-m9-mstd0.5-inc1 | rand-m9-mstd0.5-inc1 | rand-m9-mstd0.5-inc1 |
| Mixup | 0.8 | 0 | 0 |
| Cutmix | 1.0 | 0 | 0 |
| Label smoothing | 0.1 | 0.1 | 0.1 |

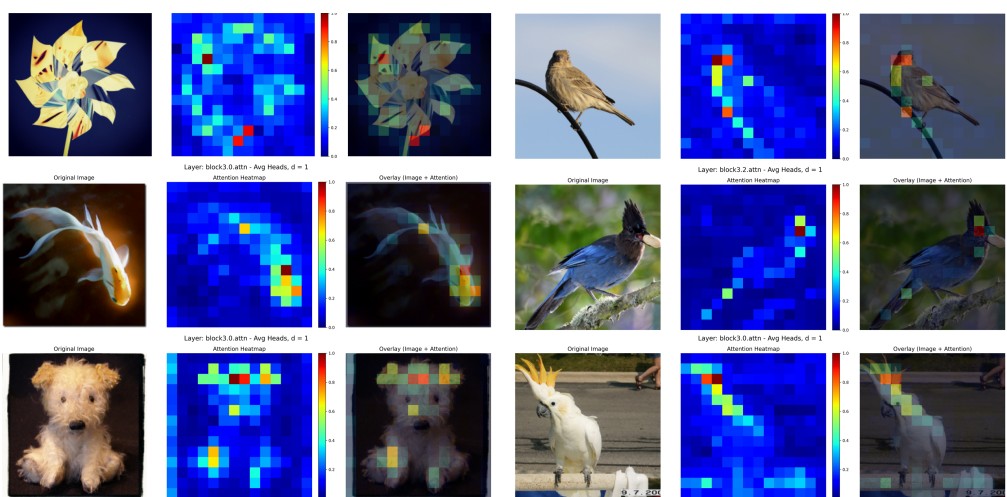

Figure 6: Attention heatmap for BSA(Ours) with Spikingformer [67] architecture on ImageNet-1K.

# E    Semantic Segmentation

This study employs the ADE20K semantic segmentation dataset [65, 66, 69, 63], comprising over 20,000 training and 2,000 validation scene-centric images with meticulous pixel-level annotations of objects and their constituent parts. The dataset exhibits rich semantic diversity, encompassing 150 semantic categories spanning environmental elements (sky, road, grass) and discrete entities (persons, vehicles, furniture).

For our experimental architecture, we utilize a SDT-V3 [52] pretrained on ImageNet-1K as the backbone network, integrated with PVT (Pyramid Vision Transformer) [46, 39] for semantic segmentation tasks. Newly introduced parameters are initialized using Xavier method. During model training, we configure a batch size of 20 with 160,000 total iterations. Our optimization strategy implements AdamW optimizer with an initial learning rate of $1 \times 10^{-4}$ and polynomial decay with power 0.9. Notably, we apply linear decay warmup during the initial 150,000 iterations to enhance model stability. Finally, our work demonstrates significant performance improvements over the vanilla V3 architecture at both 19m and 10m metrics. Additionally, we visualized the qualitative results of our model on the ADE20K dataset. As shown in Fig.7, our segmentation results exhibit remarkable effectiveness, with precise boundary delineation and enhanced semantic coherence across diverse scene contexts.

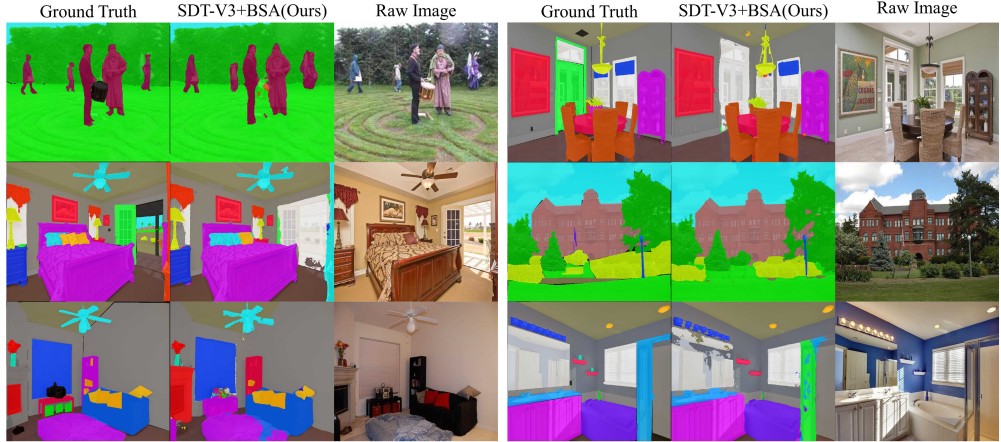

Figure 7: Visualization of semantic segmentation results on the ADE20K dataset.

## F  Event-based Tracking

We propose an efficient event-based visual tracker built upon the SDTrack pipeline, utilizing Image Pair Matching to localize targets by comparing visual features between reference and current frames. Specifically, we introduce a Hanning Window Penalty mechanism tailored for the FE108 dataset to mitigate bounding-box drift and enhance tracking stability. Dataset details include: **FE108** [58]: 108 event sequences (3,000–5,000 frames each), annotated bounding boxes, diverse scenes, optimized for high-speed tracking; **VisEvent** [40]: 60 RGB-event synchronized sequences ($\sim$250,000 frames total), varied indoor/outdoor conditions, cross-modal robustness benchmark; **FELT** [41]: 200 sequences, event accumulation in ultra-short windows (1–5 ms), challenging real-time scenarios (motion blur, occlusion, rapid motion). The training procedure includes 100 epochs for FE108 and VisEvent, and 300 epochs for FELT. Random sampling per epoch selects 60k image pairs (maximum interval of 200 frames) for FE108 and FELT, and 30k pairs for VisEvent, ensuring sample diversity. Optimization employs AdamW (initial LR=$4 \times 10^{-4}$, decayed at 80% epochs to $4 \times 10^{-5}$, weight decay=$1 \times 10^{-4}$).

## G  Limitations

The limitations of this study include performance testing of BSA on larger model sizes such as LLaMA (7B, 16B and 70B) and the deployment challenges related to the shift in hardware. These issues will be addressed in future research. The experimental results presented in this paper are reproducible. Detailed explanations of model training and configuration are provided in the main text and supplemented in the appendix. Our code and models will be made available on GitHub after the paper is accepted. In the future, we will deploy BSA onto hardware platforms such as Field Programmable Gate Arrays (FPGAs) to evaluate its practical performance. During this process, we will optimize the appropriate read-write data streams and memory access schemes to enhance the inference speed of the model.

