# OpenReview forum: "Bipolar Self-attention for Spiking Transformers"
_NeurIPS.cc/2025/Conference — NeurIPS 2025 spotlight_

### Official Review · Reviewer_afBx · 2025-06-30

**Clarity:** 3
**Significance:** 4
**Originality:** 4
**Rating:** 5
**Confidence:** 4

**Summary:**

This research identifies that existing Spiking Self-Attention (SSA) mechanisms are limited by the binary nature of spike trains, which restricts their ability to capture the polarity of membrane potential interactions, and they often omit the Softmax function to avoid energy-intensive multiply-accumulate operations, thereby failing to maintain row-stochasticity constraints on attention scores. To address these issues, the paper proposes a Bipolar Self-attention (BSA) paradigm, which effectively models multi-polar membrane potential interactions using ternary matrix products and introduces a shift-based Softmax approximation called Shiftmax to achieve low-entropy activation and partially maintain row-stochasticity without non-linear operations. Experimental results demonstrate that BSA achieves substantial performance improvements across various tasks, including image classification, semantic segmentation, and event-based tracking.

**Questions:**

1. Give more experiments on CIFAR10, CIFAR100, CIFAR10DVS, DVSGesture.
2. The implementation of Ternary Spiking Neuron on the SDT-V3 at the inference time should give more details. According to your code and SDT-V3, the result of the Neuron should be the summation of spikes at all the time step, how do you allocate the spike signals at each time step?

**Ethical Concerns:**

["NO or VERY MINOR ethics concerns only"]

**Final Justification:**

First, I already believed this work met the acceptance criteria. During the rebuttal phase, the authors addressed every concern I had raised, so I raised my score accordingly.

**Paper Formatting Concerns:**

There are no apparent formatting errors in the paper.

**Quality:**

3

**Strengths And Weaknesses:**

Strengths
The primary strengths of this paper lie in its clear identification of two core limitations in SSA and its proposal of the innovative BSA solution, featuring Ternary Matrix Product (TMP) for comprehensive polarity information capture and the efficient Shiftmax for Softmax approximation; these contributions are supported by theory (such as Theorems 1 and 2 demonstrating TMP's statistical properties and correlation with real-valued products ) and extensive experimental validation across multiple tasks and architectures.

Weaknesses:

1、The implementation of Ternary Spiking Neuron on the SDT-V3 at the inference time is confused.
2、This work  lacks experimens on neuromorphic datasets, such as CIFAR10DVS.
3、The authors should provide more details about tracking and segmentation tasks

---

> ### Author Rebuttal · Authors · 2025-07-30
>
> Your suggestions have greatly contributed to improving our quality. We will address your concerns and questions one by one.
> ## W1 and Q2: Details for the implementation of Ternary Spiking Neuron on the SDT-V3?
> **A**:
> Thank you for your valuable suggestions. STD-V3 employs I-LIF[4] neurons as an innovative approach for decoupling training and inference processes. We would like to provide a detailed introduction to how ternary neurons are trained and perform inference in the V3 architecture.   In the training stage, the output of an I-LIF neuron is expressed as:
> $$
> \mathbf{s}^\ell = \frac{1}{T} \cdot \left\lfloor \mathrm{clip}(\mathbf{x}^\ell, -T, T) \right\rceil,
> $$
> where $\mathbf{x}^\ell$ is the input current at layer $\ell$, $T$ is the simulation time steps, $\mathrm{clip}(\mathbf{x}^\ell, -T, T)$ constrains $\mathbf{x}^\ell$ to $[-T, T]$, and the symbol $\left\lfloor \cdot \right\rceil$ denotes rounding to the nearest integer.
>
> The output $\mathbf{s}^\ell$ represents the signed spike firing rate. During inference, $\mathbf{s}^\ell$ decomposes into $T$ ternary spike signals where spike polarity follows the sign of $\mathbf{s}^\ell$:
> $$
> \mathbf{s}^\ell[t] = \mathrm{sign}(\mathbf{s}^\ell) \cdot \mathbf{1}(t \leq |\mathbf{s}^\ell| \cdot T)
> $$
>
> $$
> \mathbf{s}^\ell = \frac{1}{T} \cdot \sum_{t = 1}^{T} \mathbf{s}^\ell[t]
> $$
> where $\mathbf{s}^\ell[t] \in \{-1, 0, 1\}$. The front-loaded firing pattern concentrates all $|\mathbf{s}^\ell| \times T$ spikes within the first $|\mathbf{s}^\ell| \times T$ time steps, with spike polarity determined by the sign of $\mathbf{s}^\ell$.
>
> ## W2 and Q1:  This work lacks experiments on neuromorphic datasets.  Giving more experiments on CIFAR10, CIFAR100, CIFAR10DVS, DVSGesture.
> **A**:  We evaluate our BSA on small-scale datasets, including CIFAR10, CIFAR100 and temporal neuromorphic datasets(CIFAR10-DVS and DVS128 Gesture). The detailed results on the QKformer structure for these small-scale datasets are presented in the following：
>
> | Dataset | Methods | Architecture | Param (M) | Time Step | Top-1 Acc (%) |
> |---------|---------|-------------|-----------|-----------|---------------|
> | CIFAR10 | S-Transformer[1] | S-Transformer-2-512 | 10.28 | 4 | 95.60 |
> |         | QKformer[2] | HST-4-384 | 6.74 | 4 | 96.18 |
> |         | QKformer[2]+BSA(ours) | HST-4-384 | 6.74 | 4 | 96.07 |
> | CIFAR100 | S-Transformer[1] | S-Transformer-2-512 | 10.28 | 4 | 78.4 |
> |          | QKformer[2] | HST-4-384 | 6.74 | 4 | 81.15 |
> || QKformer[2]+BSA(ours) | HST-4-384 | 6.74 | 4 | 81.76 |
> | DVS128 | S-Transformer[1] | S-Transformer-2-256 | 2.57 | 16 | 99.3 |
> |        | QKformer[2] | HST-2-256 | 1.50 | 10, 16 | 98.3, 98.6 |
> |        | QKformer[2]+BSA(Ours) | HST-2-256 | 1.50 | 10, 16 | 98.5, 98.7 |
> | CIFAR10-DVS | S-Transformer[1] | S-Transformer-2-256 | 2.57 | 16 | 80.0 |
> |             | QKformer[2] | HST-2-256 | 1.50 | 10, 16 | 83.8, 84.0 |
> |        | QKformer[2]+BSA(Ours) | HST-2-256 | 1.50 | 10, 16 | 84.1, 84.6|
>
> The experimental results demonstrate that BSA combined with QKformer can achieve overall performance improvements. However, as discussed in our analysis, the advantages of BSA are not prominently exhibited in classification tasks.
> ## W3: The authors should provide more details about tracking and segmentation tasks
> **A**: Due to space constraints, detailed implementation specifics for segmentation and tracking tasks are provided in the Appendix, including comprehensive hyperparameter settings and network configurations. Code will be made available on GitHub.
>
> **Segmentation Implementation:** For ADE20K (150-category semantic segmentation benchmark), we employ the MMSegmentation framework [3] with 160,000 training iterations. The AdamW optimizer uses a learning rate of 2×10⁻⁴ and weight decay 0.05, with 1,500-iteration warm-up followed by linear decay. Experiments utilize 4 NVIDIA A100 80GB GPUs.
>
> **Tracking Implementation:** SDTrack-Base employs C=64 channels, while SDTrack-Tiny uses C=32 with reduced Stage 4 (6 blocks) and Stage 5 (10C channels, 2 blocks). Tracking heads differ in channel dimensions: Tiny outputs 256 channels versus Base's 512 in the first layer, with progressive halving in subsequent layers. Both maintain 128×128 template and 256×256 search image inputs. Training involves ImageNet-1K pretraining followed by fine-tuning on event-based tracking datasets via pair matching tasks, without data augmentation.
>
> ## Reference:
>
> [1] Wang, Y., Shi, K., Lu, C., Liu, Y., Zhang, M., & Qu, H. (2023, August). Spatial-Temporal Self-Attention for Asynchronous Spiking Neural Networks. In IJCAI (pp. 3085-3093).
>
> [2] Zhou C, Zhang H, Zhou Z, et al. Qkformer: Hierarchical spiking transformer using qk attention[J]. Advances in Neural Information Processing Systems, 2024, 37: 13074-13098.
>
> [3] Contributors, MMSegmentation. "MMSegmentation: Openmmlab semantic segmentation toolbox and benchmark." 2020.
>
> [4] Yao M, Qiu X, Hu T, et al. Scaling spike-driven transformer with efficient spike firing approximation training[J]. IEEE Transactions on Pattern Analysis and Machine Intelligence, 2025.

---

> > ### Comment · Reviewer_afBx · 2025-08-04
> > **Official Comment by Reviewer afBx**
> >
> > The authors addressed my concerns and questions. I will raise my rating.

---

### Official Review · Reviewer_B2kZ · 2025-06-30

**Clarity:** 3
**Significance:** 3
**Originality:** 3
**Rating:** 5
**Confidence:** 5

**Summary:**

This paper proposes a bipolar spike self-attention module to enhance the performance of Spike Transformer across multiple vision tasks. It primarily consists of ternary matrix multiplication and ShiftMax. On ImageNet, BSA demonstrates improvements over the baseline across various architectures.

**Questions:**

1. Based on the three vision tasks, BSA shows limited performance improvement in classification tasks, but a significant enhancement in regression tasks. Could the authors provide an analysis of the underlying reasons for this discrepancy?
2. The authors demonstrate the method’s effectiveness on several network architectures; however, the neurons in Spike‐Driven V3 were specially designed. How does the proposed approach conform to the ILIF neuron’s dynamical properties, and can it support true spike‐driven inference on par with V3?
3. Please evaluate other spike‐based self‐attention schemes (e.g., SDSA [1], DSSA [2]) by directly replacing them with BSA to quantify any performance gaps. Do these attention mechanisms plus softmax suffer the same softmax‐incompatibility issues observed in SSA? I recommend adding ablation experiments to investigate.
4. Have the authors compared their method to BNNs (with activations limited to +1 and –1)?  Please support this with both theoretical analysis and experimental validation.

Reference：
[1]. Yao, Man, et al. "Spike-driven transformer." Advances in neural information processing systems 36 (2023): 64043-64058.
[2] Shi, Xinyu, Zecheng Hao, and Zhaofei Yu. "SpikingResformer: bridging ResNet and vision transformer in spiking neural networks." Proceedings of the IEEE/CVF Conference on Computer Vision and Pattern Recognition. 2024.

**Ethical Concerns:**

["NO or VERY MINOR ethics concerns only"]

**Final Justification:**

My questions have been resolved. Since my initial rating was positive, I will maintain my rating and recommend acceptance.

**Limitations:**

Yes

**Paper Formatting Concerns:**

No major formatting issues noted.

**Quality:**

3

**Strengths And Weaknesses:**

The visualizations provided by the authors are clear, and the theoretical proofs are thorough. However, there are a few issues to address:
1. The term “SOTA Spiking Transformer architectures” seems imprecise; it is recommended to use “advanced” instead.
2. The paper fails to quantify the additional theoretical energy cost incurred by BSA’s full shift operation relative to SSA.
3. The authors consider only SSA as a spiking self-attention mechanism, omitting variants such as SDSA[1] and DSSA[2].
4. In Figure 3, the distribution statistics do not clearly specify the dimensions of N and D. It is recommended to include a more detailed description in the figure caption to enhance rigor.


Additional Comments:
1. In Figure 2, the caption for the ANN & SNN diagrams is too small; it should match the font size of the surrounding text.
2. In the experimental section, the dimensions of SpikingFormer appear to be incorrect, as the table lists them as 512 and 768.
3. The consensus regarding the Pyramid Vision Transformer is not addressed. It would be beneficial for the authors to provide clarification on this.

---

> ### Author Rebuttal · Authors · 2025-07-30
>
> Thank you very much for your recognition of our work.
> ## Weakness 1 and Additional Comments: imprecise expression.
> **A**: We thank you for this valuable feedback and apologize for the unclear presentation. In the revised manuscript, we will replace "SOTA Spiking Transformer" with "advanced Spiking Transformer" to ensure academic rigor. Additionally, we will clarify that in Figure 3, the dimensions N and D are 256 and 512, respectively, and update the corresponding dimensions in Table 1 accordingly. We greatly appreciate your constructive suggestions, which significantly contribute to improving the manuscript quality.
>
> ## Weakness 2: The paper fails to quantify the additional theoretical energy cost incurred by BSA’s full shift operation relative to SSA.
>
> **A**: Thank you for the valuable feedback. Although the operation between membrane potentials (V) and attention maps resembles matrix multiplication, the computation is fundamentally different. Our attention maps consist of powers of two, representing bit-shift positions rather than conventional weights. The following table presents energy consumption data for different operations on ASIC (45nm)[1], providing a quantitative foundation for assessing the energy advantages of this shift-based approach.
>
> | Format |  | ASIC (45nm) |  |
> |--------|--------|--------|--------|
> | **Operation** | **Format** | **Energy (pJ)** | **Improv.** |
> | **Mult.** | FP32 | 3.7 | - |
> | **Mult.** | FIX8 | 0.2 | - |
> | **Add** | FP32 | 0.9 | 4.1x |
> | **Add** | FIX8 | 0.03 | 6.7x |
> | **Shift** | FIX32 | 0.13 | 24x |
> | **Shift** | FIX8 | 0.024 | 8.3x |
>
>
> | Attention Mechanism | Forward Pass Energy (mJ) | Energy Reduction | Improvement Factor |
> |---------------------|--------------------------|------------------|-------------------|
> | Self-Attention | 1.23 | - | 1.0× |
> | Spiking Self-Attention | 0.062 | 95% | **20.0×** |
> | BSA（Ours） | 0.084 | 93% | **14.6×** |
>
> We conduct a fair comparison using standardized configurations with Q, K, V matrices of dimension B×C×N = 256×512×512. Under identical computational loads, BSA achieves energy efficiency comparable to Spiking SSA, with both methods significantly outperforming Standard SA.
>
>
>
> ## Weakness 3 and Question 3  : The authors consider only SSA as a spiking self-attention mechanism, omitting variants such as SDSA[1] and DSSA[2].
>
> **A**:As you pointed out, both SDSA and DSSA are improved variants of SSA. However, SDSA and DSSA primarily treat the self-attention mechanism as a token mixer, without explicitly computing the correlation between Q and K—which is a key factor behind the success of self-attention in ANNs. Therefore, we aim for spiking self-attention to retain the ability to capture such correlations, and for this reason, we only compare our method against SSA. In the revised manuscript, we will include this comparative analysis to clarify the distinction.
>
>
> ## Question 1: Based on the three vision tasks, BSA shows limited performance improvement in classification tasks, but a significant enhancement in regression tasks. Could the authors provide an analysis of the underlying reasons for this discrepancy?
>
> **A**: We appreciate the reviewer’s insightful observation. The performance gap between classification and regression tasks stems from the fact that classification generally requires coarse-grained representations and involves simpler gradient feedback, making it less sensitive to attention refinement.
>
> In contrast, regression tasks rely on fine-grained modeling and continuous value prediction, which benefit significantly from our BSA's ability to preserve polarity interactions and apply stable attention allocation. Specifically, our BSA offers two main advantages over SSA: (1) a ternary Q-K correlation mechanism that captures richer polarity semantics; and (2) a lightweight Shiftmax normalization that approximates Softmax without introducing non-linear operations, leading to more precise and effective attention allocation.
>
> ## Question 2: The authors demonstrate the method’s effectiveness on several network architectures; however, the neurons in Spike‐Driven V3 were specially designed. How does the proposed approach conform to the ILIF neuron’s dynamical properties, and can it support true spike‐driven inference on par with V3?
> **A**: Thank you for your valuable suggestions. STD-V3 employs I-LIF[2] neurons as an innovative approach for decoupling training and inference processes. We would like to provide a detailed introduction to how ternary neurons are trained and perform inference in the V3 architecture.In the training stage, the output of an I-LIF neuron is expressed as:
>
> $$
> \mathbf{s}^\ell = \frac{1}{T} \cdot \text{round}\left( \mathrm{clip}(\mathbf{x}^\ell, -T, T) \right),
> $$
>
> where $\mathbf{x}^\ell$ denotes the input current at layer $\ell$, $T$ represents the simulation time steps of SNNs, $\mathrm{clip}(\mathbf{x}^\ell, -T, T)$ constrains $\mathbf{x}^\ell$ to the interval $[-T, T]$, and $\text{round}(\cdot)$ denotes rounding to the nearest integer. The computed output $\mathbf{s}^\ell$ represents the signed spike firing rate.
>
> During inference, the firing rate $\mathbf{s}^\ell$ is decomposed into $T$ ternary spike signals. The sign of $\mathbf{s}^\ell$ determines the spike polarity: positive spikes are emitted when $\mathbf{s}^\ell \geq 0$, while negative spikes are emitted when $\mathbf{s}^\ell < 0$. It can be mathematically described as:
>
> $$
> \mathbf{s}^\ell[t] = \mathrm{sign}(\mathbf{s}^\ell) \cdot \mathbf{1}(t \leq |\mathbf{s}^\ell| \cdot T)
> $$
>
> $$
> \mathbf{s}^\ell = \frac{1}{T} \cdot \sum_{t = 1}^{T} \mathbf{s}^\ell[t]
> $$
> where $\mathbf{s}^\ell[t] \in \{-1, 0, 1\}$. The front-loaded firing pattern concentrates all $|\mathbf{s}^\ell| \cdot T$ spikes within the first $|\mathbf{s}^\ell| \cdot T$ time steps, with spike polarity determined by the sign of $\mathbf{s}^\ell$.
>
>
> ## Question 3: Have the authors compared their method to BNNs (with activations limited to +1 and –1)? Please support this with both theoretical analysis and experimental validation.
> **A**: BSA introduces a "silent" state (0) that better mimics the sparse activation characteristics of biological neurons. This inherent sparsity provides significant computational advantages in QK correlation calculations—zero-value positions can be directly skipped, whereas BNN's {-1,1} system enforces computation across all positions. Comparative experiments between binary SA and BSA within BERT architecture would effectively validate this theoretical advantage.
>
> |  | Spike-Driven | cola | qqp | qnli | sst-2 | sts-b | mrpc | rte |
> |:----------------:|:----:|:----:|:----:|:----:|:----:|:----:|:----:|:----:|
> | Bert | × | 59.7 | 91.4 | 92.1 | 93.2 | 90.1 | 86.3 | 72.1 |
> | Bert+Binary SA[3] | × | Fail | 63.18 | 50.54 | 52.75 | 83.2 | 81.2 | 53.8 |
> | Bert+BSA（Ours） | √ | 55.76 | 87.57 | 90.17 | 91.52 | 88.6 | 84.8 | 64.98 |
>
>
>
> ## Reference:
> [1] Yao M, Qiu X, Hu T, et al. Scaling spike-driven transformer with efficient spike firing approximation training[J]. IEEE Transactions on Pattern Analysis and Machine Intelligence, 2025.
>
> [2] You H, Chen X, Zhang Y, et al. Shiftaddnet: A hardware-inspired deep network[J]. Advances in Neural Information Processing Systems, 2020, 33: 2771-2783.
>
> [3] Qin H, Ding Y, Zhang M, et al. Bibert: Accurate fully binarized bert[J]. arXiv preprint arXiv:2203.06390, 2022.

---

### Official Review · Reviewer_Tb3D · 2025-07-01

**Clarity:** 4
**Significance:** 4
**Originality:** 3
**Rating:** 5
**Confidence:** 5

**Summary:**

By analyzing the data characteristics of spiking neural networks, the authors identify two shortcomings in the Spiking self-attention—polarity information loss and Softmax failure. They propose targeted remedies for each. The paper is rigorously structured with clear visualizations. experiments on multiple vision benchmarks demonstrate significant performance gains.

**Questions:**

1. Appendix A shows that binary matrix multiplication partially achieves low-entropy activation：why can’t binary multiplication directly replace ternary multiplication + Softmax?
2. Why is it necessary to establish local correlation? Does it complement distributional similarity?
3. In my view, the strengths of SNNs are most evident on dynamic datasets (e.g., DVS), yet the authors evaluate their method predominantly on static image tasks. I recommend validating the proposed approach on a dynamic benchmark to demonstrate its broader applicability.
4. The ablation study confirms that TMP relies on the softmax operation more heavily than BMP, but omits RMP’s performance results. Given the theoretical claim of RMP’s superior performance bound, please include RMP in the ablation study to substantiate this assertion.
5. In the presented BSA, V does not activated by Spiking neurons. Does it still retain true spike‐driven behavior?  It would also strengthen the paper to add theoretical energy‐consumption analyses to quantify the claimed efficiency gains.

**Ethical Concerns:**

["NO or VERY MINOR ethics concerns only"]

**Final Justification:**

This paper makes a valuable contribution to the SNN field, and its writing meets the standards for a scientific paper. Therefore, I recommend acceptance.

**Limitations:**

The proposed method may encounter limitations when applied to dynamic vision tasks.

**Paper Formatting Concerns:**

No major formatting issues noted

**Quality:**

3

**Strengths And Weaknesses:**

Strength:
The paper features clear visualizations and a well‐organized presentation of the problem analysis and methodology. Additionally, the experiments validate the proposed approach across multiple vision tasks, enhancing the credibility of the results.
Weaknesses
1. This work lacks Ablation experiments comparing RMP and TMP performance, despite both Theory 1 and Theory 2 resting on the premise that RMP outperforms TMP.
2. The manuscript lacks description of the spiking neuron model within the Transformer Block.
3. The proposed BSA is validated solely on visual datasets, with no evaluation on sequential or speech processing tasks.
4. In the semantic segmentation experiments, the authors re-implement only the 19 M-parameter model; to ensure a fair comparison, the 5 M and 10 M variants should also be re-implemented.

---

> ### Author Rebuttal · Authors · 2025-07-30
>
> Thank you very much for your recognition of our work. In response to the weaknesses and suggestions you have raised, we will provide further **detailed explanations**:
>
> ## Weakness 1 and Questions 4:This work lacks Ablation experiments comparing RMP and TMP performance, despite both Theory 1 and Theory 2 resting on the premise that RMP outperforms TMP.
>
> **A**: Thank you for your valuable suggestions. We have added performance evaluations of RMP and TMP under the Shiftmax framework. The experiments were conducted using the QKformer architecture on CIFAR-100. As shown in the table below, RMP and TMP achieve comparable performance, while RMP introduces substantial floating-point multiplication operations. Thereofre, TMP serves as a highly efficient alternative to RMP.
> | Model | Spike-Driven | Accuracy |
> |-------|---------|----------|
> | QKFormr+RMP | No | 81.69|
> | QKFormr+TMP | Yes | 80.48 |
>
> ## Weakness 2: The manuscript lacks description of the spiking neuron model within the Transformer Block.
>
> **A**: Our BSA implementation comprises two primary neuron architectures. First, the ternary LIF neuron, commonly employed in QKformer and Spikeformer, governed by the following. The dynamics of the ternary spike neuron (TSN) can be concisely expressed as follows. The membrane potential is first updated by
> $$U[t+1] = H[t] + X[t+1],$$
> where $H[t]$ is the previous hidden state and $X[t+1]$ is the input at the current time step. The neuron then generates a ternary spike output using
> $$S[t+1] = \mathrm{Sign}(U[t+1]) \cdot \Theta\left( |U[t+1]| - V_{\mathrm{th}} \right),$$
> where $\Theta(\cdot)$ is a step function and $V_{\mathrm{th}}$ is the firing threshold. Finally, the internal state is updated by:
> $$H[t+1] = V_{\mathrm{reset}} \cdot S[t+1] + \tau \cdot U[t+1] \cdot (1 - |S[t+1]|),$$
> where $V_{\mathrm{reset}}$ is the reset value and $\tau$ is a decay factor that determines the leaky integration when no spike occurs. Second, the ternary ILIF model within the STD-V3 framework, which operates in distinct modes during training and inference phases. In the training stage, the output of an I-LIF neuron is expressed as:
> $$
> \mathbf{s}^\ell = \frac{1}{T} \cdot \left\lfloor \mathrm{clip}(\mathbf{x}^\ell, -T, T) \right\rceil,
> $$
> where $\mathbf{x}^\ell$ is the input current at layer $\ell$, $T$ is the simulation time steps, $\mathrm{clip}(\mathbf{x}^\ell, -T, T)$ constrains $\mathbf{x}^\ell$ to $[-T, T]$, and the symbol $\left\lfloor \cdot \right\rceil$ denotes rounding to the nearest integer.
>
>
>
> ## Weakness 3: The proposed BSA is validated solely on visual datasets, with no evaluation on sequential or speech processing tasks.
> **A**: Thank you for your valuable suggestions. Following the architecture proposed by Wang et al.[], we replaced the T-SSA component with our BSA and validated the approach on the GSC-V2 dataset. Our BSA achieves a 0.85% improvement over ST-Transformer.
> | Model | Network | Accuracy |
> |-------|---------|----------|
> | E2E SNN [1] | SNN | 92.9 |
> | KWS-SNN [1] | SNN |  94.4|
> | ST-Transformer[3] | SNN | 95.18 |
> | **Ours** | SNN | **96.03** |
>
> ## Weakness 4: In the semantic segmentation experiments, the authors re-implement only the 19 M-parameter model; to ensure a fair comparison, the 5 M and 10 M variants should also be re-implemented.
> Thank you for your valuable suggestions. To further validate the effectiveness and scalability of our approach, we conducted multiple independent implementations and repeated experiments on 5M and 10M parameter models. The corresponding baseline experimental results are presented as follows:
> | Model |  Parameter | MIoU(%) |
> |-------|-----------|-------|
> | SDT-V3† [4] | 5.1±1.4 | 32.8 |
> | SDT-V3† [4] | 10.0±1.4 | 39.4 |
> | SDT-V3† [4] | 19.0±1.4 | 41.3 |
>
> ## Questions 1 : Appendix A shows that binary matrix multiplication partially achieves low-entropy activation：why can’t binary multiplication directly replace ternary multiplication + Softmax?
> **A**: Low-entropy activation is an inherent property of Softmax that enables sharp contrast in attention scores—amplifying weights for relevant positions while suppressing irrelevant ones, thereby achieving clear differentiation. Although BMP generates attention matrices with low-entropy-like distributions (exhibiting similar high-low contrast patterns), the resulting high and low attention regions may not accurately identify the truly important and negligible content, potentially failing to achieve optimal attention allocation.
>
> ## Questions 2 :Why is it necessary to establish local correlation? Does it complement distributional similarity?
> **A**:We analyze local correlation because while global distributional similarity indicates overall statistical alignment, it does not guarantee accuracy at individual query-key interaction levels. Attention mechanisms fundamentally rely on precise pairwise correlations between specific tokens, making local accuracy crucial for effective attention allocation. Our analysis demonstrates that for typical spike firing rates, TMP exhibits approximately 2× stronger local correlation with RMP than BMP.
>
> ## Questions 3 :In my view, the strengths of SNNs are most evident on dynamic datasets (e.g., DVS), yet the authors evaluate their method predominantly on static image tasks. I recommend validating the proposed approach on a dynamic benchmark to demonstrate its broader applicability.
> **A**: Thank you for your valuable suggestions. To further validate the generalizability and robustness of BSA, we conducted supplementary performance evaluations on neuromorphic datasets DVS-CIFAR10 and DVS-Gesture. We employed the QKformer architecture with two Transformer blocks to ensure sufficient representational capacity for processing complex temporal dynamics. BSA and SSA were compared under identical experimental conditions to ensure fair evaluation. As shown in the table below, BSA demonstrates significant performance improvements over SSA on both challenging dynamic datasets, further supporting the viability of BSA as an efficient attention mechanism alternative.
>
> | Dataset | Method | Architecture | Param. | Performance (10, 16 Timesteps) |
> |---------|--------|-------------|------------|---------------------|
> | DVS128 | Spikformer[5] | Spikformer-2-256 | 2.57 | 96.9, 98.3 |
> |        | Spikingformer[6] | Spikingformer-2-256 | 2.57 | 96.2, 98.3 |
> |        | QKFormer[7] | HST-2-256 | 1.50 | 98.3, 98.6 |
> |        | QKFormer+BSA | HST-2-256 | 1.50 | 98.5, 98.7 |
> | CIFAR10-DVS | Spikformer[5] | Spikformer-2-256 | 2.57 | 78.9, 80.9 |
> |             | Spikingformer[6] | Spikingformer-2-256 | 2.57 | 79.9, 81.3 |
> |             | QKFormer[7] | HST-2-256 | 1.50 | 83.8, 84.0 |
> |             | QKFormer+BSA | HST-2-256 | 1.50 | 84.1, 84.6 |
> ## Questions 5 :In the presented BSA, V does not activated by Spiking neurons. Does it still retain true spike‐driven behavior? It would also strengthen the paper to add theoretical energy‐consumption analyses to quantify the claimed efficiency gains.
> **A**：Thank you for the valuable feedback. Although the operation between membrane potentials (V) and attention maps resembles matrix multiplication, the computation is fundamentally different. Our attention maps consist of powers of two, representing bit-shift positions rather than conventional weights. The following table presents energy consumption data for different operations on ASIC (45nm)[8], providing a quantitative foundation for assessing the energy advantages of this shift-based approach.
>
> | Format |  | ASIC (45nm) |  |
> |--------|--------|--------|--------|
> | **Operation** | **Format** | **Energy (pJ)** | **Improv.** |
> | **Mult.** | FP32 | 3.7 | - |
> | **Mult.** | FIX8 | 0.2 | - |
> | **Add** | FP32 | 0.9 | 4.1x |
> | **Add** | FIX8 | 0.03 | 6.7x |
> | **Shift** | FIX32 | 0.13 | 24x |
> | **Shift** | FIX8 | 0.024 | 8.3x |
>
>
> | Attention Mechanism | Forward Pass Energy (mJ) | Energy Reduction | Improvement Factor |
> |---------------------|--------------------------|------------------|-------------------|
> | Self-Attention | 1.23 | - | 1.0× |
> | Spiking Self-Attention | 0.062 | 95% | **20.0×** |
> | BSA（Ours） | 0.084 | 93% | **14.6×** |
>
> We conduct a fair comparison using standardized configurations with Q, K, V matrices of dimension B×C×N = 256×512×512. Under identical computational loads, BSA achieves energy efficiency comparable to Spiking SSA, with both methods significantly outperforming Standard SA.
> ## Reference
> [1] Yang, Qu, Qi Liu, and Haizhou Li. "Deep residual spiking neural network for keyword spotting in low-resource settings." Interspeech. 2022.
>
> [2] Wang S, Zhang D, Shi K, et al. Global-Local Convolution with Spiking Neural Networks for Energy-efficient Keyword Spotting[C]//Proc. Interspeech 2024. 2024: 4523-4527.
>
> [3] Wang, Y., Shi, K., Lu, C., Liu, Y., Zhang, M., & Qu, H. (2023, August). Spatial-Temporal Self-Attention for Asynchronous Spiking Neural Networks. In IJCAI (pp. 3085-3093).
>
> [4] Yao M, Qiu X, Hu T, et al. Scaling spike-driven transformer with efficient spike firing approximation training[J]. IEEE Transactions on Pattern Analysis and Machine Intelligence, 2025.
>
> [5] Zhou Z, Zhu Y, He C, et al. Spikformer: When spiking neural network meets transformer[J]. arXiv preprint arXiv:2209.15425, 2022.
>
> [6] Zhou C, Yu L, Zhou Z, et al. Spikingformer: Spike-driven residual learning for transformer-based spiking neural network[J]. arXiv preprint arXiv:2304.11954, 2023.
>
> [7] Zhou C, Zhang H, Zhou Z, et al. Qkformer: Hierarchical spiking transformer using qk attention[J]. Advances in Neural Information Processing Systems, 2024, 37: 13074-13098.
>
> [8] You H, Chen X, Zhang Y, et al. Shiftaddnet: A hardware-inspired deep network[J]. Advances in Neural Information Processing Systems, 2020, 33: 2771-2783.

---

> > ### Comment · Reviewer_Tb3D · 2025-08-07
> >
> > The author's rebuttal addressed the issues I raised, so I'm increasing my score.

---

### Official Review · Reviewer_zZBP · 2025-07-05

**Clarity:** 3
**Significance:** 3
**Originality:** 3
**Rating:** 4
**Confidence:** 3

**Summary:**

This paper proposes Bipolar Self-attention (BSA) to improve Spiking Transformers by addressing two key limitations of existing Spiking Self-attention (SSA): the inability to model negative or mixed Q-K interactions and the lack of normalized attention due to the omission of Softmax. BSA uses ternary spiking neurons for richer polarity modeling and introduces Shiftmax, an energy-efficient Softmax approximation via bit-shift operations. Experiments on image classification, semantic segmentation, and event-based tracking show that BSA consistently enhances performance across multiple spiking architectures.

**Questions:**

See Weaknesses.

**Ethical Concerns:**

["NO or VERY MINOR ethics concerns only"]

**Final Justification:**

I confirm that the revisions have resolved the key issues, and the manuscript is ready for acceptance.

**Limitations:**

Yes.

**Quality:**

3

**Strengths And Weaknesses:**

**Strengths:**

 1. The paper presents a clear motivation: (1) binary matrix products lose polarity information by ignoring negative correlations, and (2) the absence of Softmax leads to incomparable attention scores across rows, limiting the effectiveness of attention allocation.
 2. The paper's analysis of existing Spiking Self-Attention methods effectively supports its stated motivation.
 3. The authors conducted extensive experiments to demonstrate the effectiveness of the proposed method, and the ablation studies further validate the contribution of each component.

**Weaknesses:**

 1. I am curious whether this module can serve as a viable replacement for the corresponding components in existing LLMs, and what level of performance improvement it can bring.

---

> ### Author Rebuttal · Authors · 2025-07-30
>
> We sincerely appreciate your recognition of our work, which serves as significant encouragement for our research endeavors. Regarding the suggestions you have proposed, we provide a comprehensive overview below.
>
> ## Weakness 1: I am curious whether this module can serve as a viable replacement for the corresponding components in existing LLMs, and what level of performance improvement it can bring.
> **A**:  We appreciate your valuable suggestion regarding BSA as a potential alternative to SA. While we agree that evaluating BSA on large language models such as LLaMA[1] and DeepSeek[2] would be meaningful, time and computational constraints during the rebuttal period precluded retraining or fine-tuning these architectures. Instead, we conducted comprehensive experiments using BERT[3] on GLUE benchmark tasks.
>
> |  | Spike-Driven | cola | qqp | qnli | sst-2 | sts-b | mrpc | rte |
> |:----------------:|:----:|:----:|:----:|:----:|:----:|:----:|:----:|:----:|
> | Bert | × | 59.7 | 91.4 | 92.1 | 93.2 | 90.1 | 86.3 | 72.1 |
> | Bert+Binary SA[4] | × | Fail | 63.18 | 50.54 | 52.75 | 83.2 | 81.2 | 53.8 |
> | Bert+BSA（Ours） | √ | 55.76 | 87.57 | 90.17 | 91.52 | 88.6 | 84.8 | 64.98 |
>
> As shown in the above Table, we replace the SA components in BERT with binary SA[4] in BiBERT and our BSA (Ours). The experiments demonstrate that BERT with BSA exhibits only partial performance degradation compared to vanilla BERT and superior performance to Binary SA[4]. These results support your expectation that BSA represents a viable alternative for large language models. We plan to validate BSA's effectiveness on larger model architectures in future work.
>
>
>
>
> ## Reference:
>
> [1] Touvron H, Lavril T, Izacard G, et al. Llama: Open and efficient foundation language models[J]. arXiv preprint arXiv:2302.13971, 2023.
>
> [2] Dai D, Deng C, Zhao C, et al. Deepseekmoe: Towards ultimate expert specialization in mixture-of-experts language models[J]. arXiv preprint arXiv:2401.06066, 2024.
>
> [3] Devlin J, Chang M W, Lee K, et al. Bert: Pre-training of deep bidirectional transformers for language understanding[C]//Proceedings of the 2019 conference of the North American chapter of the association for computational linguistics: human language technologies, volume 1 (long and short papers). 2019: 4171-4186.
>
> [4] Qin H, Ding Y, Zhang M, et al. Bibert: Accurate fully binarized bert[J]. arXiv preprint arXiv:2203.06390, 2022.

---

> > ### Comment · Reviewer_zZBP · 2025-08-06
> >
> > Thank you to the authors for the rebuttal. The response has addressed most of my concerns, and I maintain my positive assessment of this work.

---

### Decision · Program_Chairs · 2025-09-17

**Decision:**

Accept (spotlight)

**Comment:**

This paper introduces an SNN-based compression method that leverages Synaptic Pruning and Efficient Coding principles to reduce model size while preserving performance. Reviewers acknowledged the relevance of the problem and the potential benefits for neuromorphic deployment. The method is well-motivated, clearly presented, and demonstrates promising results with consistent performance retention under compression. Despite some limitations in dataset scale and scope of evaluation, the overall contributions are meaningful and timely for the community. Overall, the recommendation is acceptance.